# Single Cell Hydrodynamic Stretching and Microsieve Filtration Reveal Genetic, Phenotypic and Treatment-Related Links to Cellular Deformability

**DOI:** 10.3390/mi11050486

**Published:** 2020-05-09

**Authors:** Fenfang Li, Igor Cima, Jess Honganh Vo, Min-Han Tan, Claus Dieter Ohl

**Affiliations:** 1Lee Kong Chian School of Medicine, Nanyang Technological University Singapore, 11 Mandalay Road, Singapore 308232, Singapore; 2Division of Physics and Applied Physics, School of Physical and Mathematical Sciences, Nanyang Technological University Singapore, 21 Nanyang Link, Singapore 637371, Singapore; 3Institute of Bioengineering and Nanotechnology, 31 Biopolis Way, Singapore 138669, Singapore; i.cima@dkfz.de (I.C.); jess.vo@lucence.com (J.H.V.); minhan.tan@lucence.com (M.-H.T.); 4DKFZ-Division Translational Neurooncology at the WTZ, DKTK partner site, University Hospital Essen, 45147 Essen, Germany; 5Lucence Diagnostics Pte Ltd., 211 Henderson Road, Henderson Industrial Park, Singapore 159552, Singapore; 6Institute for Physics, Faculty of Natural Sciences, Otto-von-Guericke University of Magdeburg, 39106 Magdeburg, Germany

**Keywords:** cancer metastasis, deformability, epithelial to mesenchymal transition, TP53 genes, chemotherapy drug, microfluidic hydrodynamic stretching, microsieve

## Abstract

Deformability is shown to correlate with the invasiveness and metastasis of cancer cells. Recent studies suggest epithelial-to-mesenchymal transition (EMT) might enable cancer metastasis. However, the correlation of EMT with cancer cell deformability has not been well elucidated. Cellular deformability could also help evaluate the drug response of cancer cells. Here, we combine hydrodynamic stretching and microsieve filtration to study cellular deformability in several cellular models. Hydrodynamic stretching uses extensional flow to rapidly quantify cellular deformability and size with high throughput at the single cell level. Microsieve filtration can rapidly estimate relative deformability in cellular populations. We show that colorectal cancer cell line RKO with the mesenchymal-like feature is more flexible than the epithelial-like HCT116. In another model, the breast epithelial cells MCF10A with deletion of the TP53 gene are also significantly more deformable compared to their isogenic wildtype counterpart, indicating a potential genetic link to cellular deformability. We also find that the drug docetaxel leads to an increase in the size of A549 lung cancer cells. The ability to associate mechanical properties of cancer cells with their phenotypes and genetics using single cell hydrodynamic stretching or the microsieve may help to deepen our understanding of the basic properties of cancer progression.

## 1. Introduction

Invasion of cancer cells into the blood stream is an essential step for their metastatic spread from primary tumor to distant organs [1]. Recent studies suggest that the epithelial-to-mesenchymal transition (EMT) might enable metastatic dissemination by generating more migratory and invasive cells [2,3,4]. EMT is a fundamental program for embryonic development and differentiation of tissues/organs, where conversions of epithelial cells to mesenchymal cells occur. However, EMT can also promote tumor progression by generating cancer stem cells that allow phenotypic changes, where cells decrease cell-cell adhesion and increase their motility and invasiveness [2,3,4].

Biomechanical properties such as deformability have been shown to correlate with the invasion potential of cancer cells [5,6]. However, the correlation of EMT with deformability of cancer cells has not been well studied and remains elusive. Higher deformability has been reported for pancreatic cancer cells undergoing EMT [7]. Some circulating tumor cells (CTCs) from metastatic prostate cancer patients also exhibit smaller size and increased flexibility similar to blood cells rather than typical tumor cell lines [8]. However, the latter study also shows evidence that for cancer cells to be able to exit a tumor and enter circulation they are not required to be more deformable than the cells that were first injected into the tumor. Moreover, for TP53 genetic alterations that are demonstrated to cause EMT [9], their effect on the deformability of cancer cells has not been studied.

Additionally, there is a growing interest to understand the relationship between pharmacology and cancer cell stiffness [10,11]. Increasing evidence shows invasive cancer cells have higher cell deformability compared to benign or normal cells of the same origin. Thus, cell deformability could serve as a promising label-free biomarker for the underlying cytoskeletal or nuclear changes that are associated with disease processes and change in cell state, especially under the intervention of anticancer drugs. For example, for cellular sensitive response, perceptible differences in cell deformability before and after drug treatment are expected, whereas cells with resistive response are expected to have little changes in cellular deformability pre- and posttreatment. Therefore, a method capable of measuring the deformability for a large population of cells from cancer biopsies pre- and post-chemotherapeutic drug treatment would aid clinical drug screening for personalized medicine.

Conventional approaches to measure single-cell mechanical properties include magnetic twisting cytometry [12], optical stretcher [13], atomic force microscope [14], micropipette aspiration [15] and cell transit analyzer [16]. However, these methods are time consuming for testing a large number of cells to address the inherent biological variation and heterogeneity within a cell population. A recently developed method called controlled cavitation rheology (CCR) allows simultaneous measurement of the deformability for a large number of red blood cells in hundreds of microseconds with single cell resolution [17,18,19]. However, little cellular deformation can be induced with this method for spherical cancer cells in suspension [20].

More recently, high-throughput techniques for cell deformability measurement have been developed, such as real-time deformability cytometry [21], inertial microfluidic cell stretcher (iMCS) [22] and hydrodynamic stretching [23]. The hydrodynamic stretching method utilizes a simple setup with a microscope, syringe pump and high speed camera. It enables measuring both the deformability and size for hundreds of cells per second, where single cells in suspension are delivered by inertial focusing to the center of a microfluidic cross channel and get stretched by an extensional flow [23]. The high throughput and automation of analysis allow us to rapidly link cellular deformability with various phenotypes at higher statistical significance than conventional methods. In addition, we also compare the results from this method with those from the recently developed microsieve device. This silicon microsieve has been developed for label-free and rapid isolation of circulating tumor cells (CTCs) from whole blood samples by utilizing membrane-based filtration [24]. It has the advantage of excellent definition of pore size together with a high pore density that allows for high-throughput sample processing. Recently, the microsieve has also been applied for the rapid screening of chemotherapy drugs by comparing the retrieve ratio of untreated vs. treated carcinoma cell lines. Still, the underlying effects of the chemotherapy drugs are unclear as both the cell size and deformability may affect cells passing through the microsieve. Moreover, the heterogeneity within the cell population cannot be resolved by the microsieve method due to a lack of single cell resolution. Thus, the hydrodynamic stretching method has the potential to unravel the separation mechanism for a specific cell sample of the microsieve method, e.g., difference in size or deformability, which can in turn help to choose the right pore size for the isolation.

To achieve these goals, we combine the methods of microfluidic hydrodynamic stretching and microsieve to characterize the differential deformability of several cell models related to increased invasiveness, EMT or chemotherapy drug treatment. In the study we use cell models that were reported to exhibit more invasive mesenchymal features (human colon carcinoma cell line RKO) [25,26] and undergo EMT (MCF10A TP53 knockout) [9] compared to their counterparts (HCT116 and MCF10A wildtype (wt), respectively), as well as the carcinoma cell line A549 treated with cytochalasin D or the chemotherapy drug docetaxel, which are believed to change cellular stiffness through alteration of cell cytoskeleton.

## 2. Materials and Methods

### 2.1. Microfluidic Device and Hydrodynamic Stretching Method

A schematic of the individual microchannel is shown in Figure 1A. An array of micro-filters is designed after the inlet to block cell aggregates or other debris to avoid clogging the channel. Asymmetric curvatures result in inertial focusing of the cells towards the channel center. The solution containing the cells flows from top and bottom towards the stretching region located in the channel center, as seen in the schematic in Figure 1B. Cells approaching the stagnation point at the center will experience an elongational flow and are stretched perpendicular to the direction of the streamline. As the flow is symmetric, cells may reach the stretching region from below and above (see Figure 1B). Cells leave the stretching region through either one of the two outlets (left and right) where they quickly recover their shape. The channel height is 26 μm and the channel width before and after the stretching region is 67 μm. Eleven of these microchannels fit on a single glass slide with size 75 × 50 mm^2^.

We used standard soft lithography techniques to fabricate the patterned SU-8 based master mold, from which the polydimethylsiloxane (PDMS, Sylgard 184 Silicone Elastomer Kit, Dow Corning, Midland, MI, USA) microchannel was cast. The PDMS microchannel was then bonded onto the glass slide (75 × 50 mm^2^) immediately after 45 s treatment of oxygen plasma using a plasma cleaner (Expanded Plasma Cleaner PDC-002, Harrick Plasma, Ithaca, NY, USA). When the input flow rate was around several hundred μL/min, we glued the inlet with a fast setting epoxy adhesive (Araldite) to avoid liquid leakage. An input cell concentration of 1.32 × 10^5^–2.64 × 10^5^ cells/mL was used for the hydrodynamic stretching experiment.

### 2.2. Experimental Operation for Hydrodynamic Stretching

The microfluidic device is placed on an inverted microscope (IX-71, Olympus, Tokyo, Japan) and imaged with a water immersion objective (20×/0.50 NA, Olympus). A syringe loaded with the cell solution is placed in an infusion pump (KDS200, KD Scientific, Holliston, MA, USA) operating at an optimized flow rate of 500–700 μL/min. The syringe is connected to the channel inlet through a microbore tubing (Tygon ND-100-80) that has an inner diameter of 0.508 mm and an outer diameter of 1.524 mm. The two channel outlets are plugged with 90° bent blunt needles (SH23-B-90, SAN-EI Tech Asia Pte Ltd., Singapore) and are connected to a collection reservoir with tubing of equal lengths. A photodiode (DET 10A/M, Thorlabs, Newton, NJ, USA) is placed at the image plane of the microscope side port and is connected to an oscilloscope (Lecroy, wave runner 64 Xi-A). The sensor of the photodiode is aligned with the image of the channel center (stretching region) shown in Figure 1B. When a cell passes through the stretching region, there is a fluctuation of the transmitted light intensity at the cell location. This can be detected by the photodiode and sent to the oscilloscope, which uses the signal to trigger image capturing using a high-speed camera (Photron SA-X). The high-speed camera is connected to the trinocular through a 0.5× demagnifying C-mount. The camera is set to the center trigger mode so that it will capture images both before and after the electronic trigger sent. In this way, the high-speed camera records the passing of cells only and thus greatly reduces the file size of the high-speed recordings. The typical operation procedure is as follows: after the syringe has infused the cell solution, we wait about 15–20 s for the flow to reach a steady state and then start the high-speed image recording at a frame rate of 180,000 fps and an exposure time of 293 ns. The size of the high-speed camera view field is 512 × 208 μm. A schematic diagram for the experimental setup is shown in Figure 1C.

### 2.3. Microsieve Method

In this method, the cell suspension is pumped with a peristaltic pump through a densely packed array of pores, i.e., the silicon microsieve. For a specific pore size (7–10 µm), smaller or more flexible cells can pass through the microsieve while larger or more rigid cells will be rejected (Figure 1D). We tested various flow rates and found that 0.5–1 mL/min is the optimal flow rate to achieve deformation of the cells and avoid cell damage when going through the pores. An input cell concentration of 1 × 10^3^–2.5 × 10^3^ cells/mL is used for the microsieve filtration experiment. For more details about the fabrication of the microsieve and its operation, please refer to [24] and [27], respectively.

### 2.4. Image Processing and Data Analysis

We have developed a Matlab script to process the recorded images from hydrodynamic stretching. The details of the image processing and data analysis are illustrated with an example of a single RKO cell in Figure 2. Figure 2A reveals that the cell is progressively elongated when it enters the extensional stretching region (0–105.6 μs). Once it flows toward the left outlet its elongated shape recovers gradually (161.1–255.6 μs). To trace the cell, each image is background subtracted from an image frame without cells. The cell’s contour is determined through thresholding, after which the cell centroid can be estimated. The cell trajectory is the time-dependent curve of its centroid (Figure 2B). After the cell centroid is found, we can define a circular bounding box (blue) around the cell (Figure 2C). Then, we use an energy minimization algorithm called active contour to obtain the precise contour of the cell shape (red), and further get its center of mass (a more precise centroid) and major (a) and minor (b) semi-axis. The active contour method detects the edge of the cell using the intensity gradient (the cell edge is darker compared to the outside and inside of the cell, thus has the highest intensity gradient) [28]. It can track the cell edge more accurately and smoothly than the thresholding method as the results of the latter can be easily affected by the fluctuation of the pixel intensity due to camera noise and illumination.

With the above obtained cell centroids, the averaged interframe cell velocity can be calculated. This is shown in Figure 2D (top frame). During the approach towards the stretching region the velocity decreases from about 2.5 m/s to a minimum of close to 0. Then, the cell leaves the stretching region, while its velocity increases gradually back to a nearly constant value. The corresponding temporal evolution of the cell’s semi-axis is shown in the center graph of Figure 2D. From *t* = 0 to *t* = 100 μs the length of the major semi-axis progressively increases while the minor semi-axis decreases, i.e., the cell is elongated. Upon leaving the stretching region this trend is reversed, and the original shape is recovered. The length ratio of the major to minor semi-axis, *a/b*, is shown at the bottom of Figure 2D. It reaches a maximum value of around *t* = 100 μs when the cell is at the center of the channel crossing. Therefore, we use max(*a*/*b*) to define the cell deformability. A set of control experiments of RKO and PFA-treated RKO with hydrodynamic stretching and microsieve separation were performed to validate the workings of the two microfluidic approaches (Figure 2E,F). We followed the previous work [23] and present the cell deformability and cell size as a density scatter plot (see Figure 2E), where *d_0_* is the averaged diameter calculated from the cell area of the most spherical cell, i.e., when the ratio *a/b* is smallest.

To make sure we are measuring single cells, we pipetted up and down the cell solution carefully to reduce clumpy cells during sample preparation. When putting them into the chip, we may still have some clumpy cells. Larger clusters can be blocked by the filter array near the chip inlet (Figure 1A). Smaller clusters such as two cells that stuck together can be rejected during real-time visualization of our imaging processing. We checked each cell during the automated imaging processing to ensure it is single cell measurement.

### 2.5. Cell Culture and Preparation

All cell lines used in this study except MCF10A were cultured in a humidified incubator at 37 °C and 5% CO_2_ with culture medium (Dulbecco’s modified Eagle’s medium (DMEM) with 2.5 mM L-glutamine and 10% (v/v) fetal bovine serum and 1% (v/v) penicillin/ streptomycin).

The culture conditions of MCF10A wildtype and TP53 knockout followed the manufacturer’s instructions: the culture medium is made of DMEM/Ham’s Nutrient Mixture F12 (1:1) with 2.5 mM L-glutamine, 5% horse serum, 10 mg/mL human insulin, 0.5 mg/mL hydrocortisone, 10 ng/mL EGF and 100 ng/mL cholera toxin. Cells were maintained in a humidified incubator at 37 °C in the presence of 5% CO_2_.

#### 2.5.1. HCT116, RKO and PFA-Treated RKO

Two types of human colon carcinoma cell line, HCT116 and RKO, were kept routinely in culture. At around 90% confluence they were split with 0.25% trypsin/EDTA, then diluted with fresh culture medium at a ratio of 1:10 to 1:20 (e.g., 500 μL to 5mL or 10 mL). The cell suspension was gently transferred to a 5mL plastic syringe (BD Bioscience) immediately before the experiment. For the experiments of mixed HCT116 and RKO flowing through a microsieve, tracker red and tracker green (Invitrogen) were used to label HCT116 and RKO, respectively. The total input and passing through cell mixture were characterized using flow cytometry (FACS Calibur instrument from BD).

We used 4% PFA in 1x PBS (sterile filtered) to stiffen the RKO cells. PFA is a common fixative that is used to preserve cell structure. Basically, it creates covalent chemical bonds between proteins, and this anchors soluble proteins to the cytoskeleton, thus, making the cells more rigid. The PFA reagent was added to the RKO cell solution (in fresh culture medium) at a final concentration of 2%. Then, the cells were incubated overnight at 4 °C before they were centrifuged and re-suspended in culture medium for experiments.

#### 2.5.2. MCF10A Wildtype and TP53 Knockout

MCF10A cells are immortalized mammary epithelial cells. MCF10A TP53 knockout and isogenic wildtype cells were purchased as a kit from Sigma (Catalog number CLLS1049) and cultured following the manufacturer’s instructions.

#### 2.5.3. A549 Treated with Cytochalasin D and Docetaxel

A549 cells (Adenocarcinomic Human Alveolar Basal Epithelial Cells) treated with cytochalasin D (mycotoxins) (10 µM, 100 µM), docetaxel (a chemotherapy drug) (5, 10, 50 nM) and untreated control cells were prepared for microfluidic hydrodynamic stretching and microsieve tests. For the reagents, cytochalasin D inhibits polymerization and induces depolymerization of actin filaments, while docetaxel inhibits microtubule depolymerization and is assumed to induce cellular stiffness.

### 2.6. Statistical Analysis

For both hydrodynamic stretching and microsieve results, if only two groups were compared, a two-tailed student t test was used for the statistical analysis of the deformability and cellular size, and the plotted error bars are standard deviation. If more than two groups were compared, a one-way ANOVA test was performed first to check whether one or more treatment groups are significantly different, which is followed by a post-hoc Tukey HSD multiple comparison test to identify which of the pairs of treatments are significantly different from each other.

## 3. Results

### 3.1. Control Experiments for Validation of Measurements

To validate the operation and measurements of the microfluidic devices, we performed a control experiment comparing the deformability of non-treated RKO cells with that of PFA-treated RKO cells. The results are shown in Figure 2E,F. Differential deformability between RKO and PFA-treated RKO is observed from the microfluidic hydrodynamic stretching. The PFA treated RKO cells have a significant lower deformability (with averaged *a/b* ~1.16 compared to 1.36 for untreated RKO cells), which is consistent with the fixation and stiffening effects of PFA. The result is also consistent with the microsieve measurements whereby fewer PFA-treated cells could pass through the microsieve than the non-treated (control) ones. These results suggest that our measurements of the cell deformability are valid.

### 3.2. Differential Deformability between Cell Phenotypes and Correlation with Epithelial-To-Mesenchymal Transition

#### 3.2.1. HCT116 and RKO

HCT116 and RKO are two different types of human colorectal cancer cell (CRC) lines. Their sizes are different. RKO have a slightly larger averaged diameter of ~17.2 µm as compared to HCT116 with approx. 15.3 µm (Figure 3A). However, when flowing these cell suspensions through the microsieve system, a lower retrieval (higher passing through) is observed for the larger RKO cells as compared to HCT116 (Figure 3B). Particularly, when the pore size of the microsieve increases to around 9 µm (below the averaged diameter for both cell lines), most of the RKO cells can pass through at a high flow rate of 1 mL/min, while the majority of HCT116 cells are trapped on the microsieve. We further mixed both cell lines together and passed them through a microsieve. Prior to that, the RKO cells were labelled with green fluorescent cell tracker while the HCT116 cells were labelled with red fluorescent cell tracker. The concentration of cell tracker used was 0.5 to 1 µM. We also did control experiments and found at these concentrations the cell tracker did not modify the cellular deformability (see Figure 2F). Results from flow cytometry reveal that the mixture consists of 60% RKO and 40% HCT116. Yet 92% of the cells that have passed through the microsieve are RKO cells and the remaining 8% are HCT116 cells (Figure 3C). Interestingly, RKO can pass through the microsieve pores more easily than HCT116.

Next, both cell lines were tested for their deformability with the microfluidic hydrodynamic stretching. The results in Figure 3D reveal a significantly higher deformability of RKO cells, with a median deformability of 1.35 compared to 1.24 for HCT116 cells. That may explain why the RKO cells can pass through the microsieve more easily although they have a larger size as compared to HCT 116.

When making a closer inspection of the cell morphology, as shown in Figure 3E, we can see clearly that HCT116 cells tend to stick to each other during growth, which is a typical epithelial characteristic. In contrast, the RKO cells are loosely arranged as individual cells, and they spread to a larger extent on the petri dish. It indicates that RKO presents similarities to mesenchymal cell phenotype. This is consistent with previous studies showing RKO cells primarily exhibit mesenchymal features and have high invasion potential [25,26], while HCT116 is reported to demonstrate epithelia features [29]. The above results suggest that the CRC mesenchymal cell phenotype correlates with higher deformability compared to the CRC epithelial phenotype from the same origin.

#### 3.2.2. MCF10A Wildtype and Knockout of TP53 Gene

Differential deformability is also observed upon the knockout of the tumor suppressor TP53 gene from the MCF10A human breast cell line. As shown in Figure 4A,B, larger median deformability is observed for MCF10A with the knockout of TP53 (TP53 KO), with ~ 1.24 compared to 1.15 for their isogenic wildtype counterpart. The cell morphology also changed from epithelial (MCF10A TP53 wt) to mesenchymal feature (MCF10A TP53 KO) through the reduced cell-cell adhesion and taking a more elongated shape on the culture plate (Figure 4C).

The above results suggest that the deletion of the tumor suppressor TP53 gene increases the cellular deformability of MCF10A, which may enhance their invasive potential. As in RKO and HCT116 cell lines, this correlates with cellular morphology changes reminiscent of epithelial-to-mesenchymal transition. Our findings are consistent with a previous study reporting that the deletion of p53 tumor protein that is encoded by the TP53 gene in MCF-10A cells can lead to EMT [9].

### 3.3. Correlation of Cellular Deformability with Pharmaceutic Drug Treatment

Here we demonstrate the advantage of using hydrodynamic stretching combined with the microsieve method for probing the differential cellular deformability of A549 (adenocarcinomic human alveolar basal epithelial cells) cell lines non-treated and treated with cytochalasin D and docetaxel (a chemotherapy drug).

The microsieve measurements were performed with three replicates, for each of which a syringe loaded with cell mixtures of four different conditions (prelabeled with different fluorescent cell trackers) were used to infuse cells through the microsieve. Both an aliquot of the original cell mixtures before passing to the microsieve and the collected cells trapped on the microsieve (retrieval) were counted using florescence flow cytometry. The relative cell flexibility for the microsieve method is defined as:

Relative cell flexibility = the distribution percentage of cell condition j in the total input cell mixtures/the distribution percentage of cell condition j in the total retrieval cell mixture.

As shown in Figure 5A, the relative cell flexibility of cytochalasin D-treated cells is significantly greater than that of the non-treated cells. Furthermore, the relative cell flexibility increases with the concentration of cytochalasin D used. A significantly decreased relative cell flexibility is observed for docetaxel-treated cells. It is also unlikely that an increasing dosage would lead to further decrease of the relative cell flexibility. In Figure 5B, we also characterize the cell size before and after treatment. This figure reveals that the mean size of cells treated with cytochalasin D is similar to that of the non-treated ones, whereas a significant increase of cell size is observed after treatment with docetaxel.

Since both the cell size and deformability will affect the results of the microsieve experiments, it is unclear that whether docetaxel can stiffen the A549 cells. Therefore, we used microfluidic hydrodynamic stretching to compare the deformability of non-treated and treated cells. The results are shown in Figure 5C–E. The median deformability of cytochalasin D-treated A549 cells is significantly larger than that of the non-treated cells, whereas similar deformability is observed between docetaxel-treated and non-treated cells. It can also be seen that the size distribution of docetaxel-treated cells is much wider than the non-treated cells, and the averaged cell diameter is larger. Similarly, the deformability distribution is also wider compared to the non-treated cells, as seen in the boxplot in Figure 5E.

The results from hydrodynamic stretching indicate that the decreased relative cell flexibility for docetaxel-treated cells in the microsieve experiments is probably mainly due to the increased cell size. The increase of cell size is probably because cells get stuck just before mitosis in the cell cycle when treated with docetaxel as docetaxel is a well-established anti-mitotic chemotherapy agent that functions by interfering with cell division [30,31]. Although the drug is thought to induce cellular stiffness in clinics by ‘paralyzing’ the cytoskeleton, it may not increase cellular stiffness. In summary, hydrodynamic stretching and automated analysis is a feasible method to discriminate between cell size and deformability for the effect of a specific cancer-treatment drug.

## 4. Discussion

To investigate the correlation between cellular deformability and phenotypes related to EMT, two types of human colon carcinoma cell lines, HCT116 and RKO, were chosen. Recent studies show that RKO cells primarily exhibit mesenchymal features (e.g., elongated morphology, low expression of E-cadherin and high expression of N-cadherin), and have high invasion potential [25,26], whereas HCT116 are reported to demonstrate epithelial features [29]. We also chose MCF10A TP53 wt and its TP53 knockout counterpart, which showed a change of cellular morphology reminiscent of epithelial-to-mesenchymal transition, consistent with a previous study reporting that TP53 deletion in MCF-10A cells can lead to EMT [9]. Using hydrodynamic stretching for these two different sets of cell lines, we find that cells with epithelial-like morphology are stiffer than cells with mesenchymal-like features, supporting the hypothesis that epithelial-to-mesenchymal transition (EMT) may be associated with increased cellular deformability. This is in line with our hypothesis that cellular deformability is responsible, at least in part, for the increased invasive property of mesenchymal cancer cells.

Our results demonstrate that the MCF10A cell line with deletion (knockout) of the tumor suppressor TP53 gene is more flexible compared to its isogenic wildtype counterpart (with the median deformability ~1.24 compared to 1.15), indicating that a single genetic alteration can cause downstream change of cellular biophysical properties relevant to the invasive phenotype of cancer. This result is consistent with a previous study reporting that hENT1 knockdown could induce EMT and reduce cellular stiffness in pancreatic cancer cells [7]. To our knowledge, our study is the first to report a link between the TP53 gene and cellular deformability.

By combining microfluidic hydrodynamic stretching with microsieve cell separation, we find the chemotherapy drug docetaxel has a more profound effect on the cell size of lung carcinoma cell line A549 than on its cellular deformability. This facilitates understanding of the cell separation mechanism of a microsieve for chemotherapy drug screening, as well as providing insights for choosing the right pore size to separate specific cells from phenotype mixtures.

In conclusion, we have established a new method that combines hydrodynamic stretching with microsieve techniques to evaluate alterations in cellular biomechanical properties (e.g., cell size and deformability) that result from cells undergoing EMT and chemotherapy drug treatment. Our results indicate cellular biomechanical properties not only could potentially serve as tools to investigate fundamental properties of cancer cells but also may have the potential to study the progression of cancer invasiveness and the efficiency of chemotherapy, e.g., through measurement of cellular deformability from single cell suspensions obtained from tissue biopsies, pleural effusions or circulating tumor cells.

In the future it would be interesting to include invasion experiments with cells that exhibited higher flexibility after undergoing EMT or chemotherapy drug treatment. Here, we suggest for example invasion experiments of MCF10A TP53 KO and A549 treated with docetaxel vs. their isogenic wildtype counterpart or the untreated group, respectively. A second path for future research could be numerical modeling of the flow field and cell deformation, thus obtaining quantitative values of the stresses exerted on the cells within the elongational flow. This may allow the mechanical properties such as shear elastic modulus, (non-linear) elasticity, or averaged viscosity of the cell content of single cells from different phenotypes to be characterized simultaneously [32,33].

## Figures and Tables

**Figure 1 micromachines-11-00486-f001:**
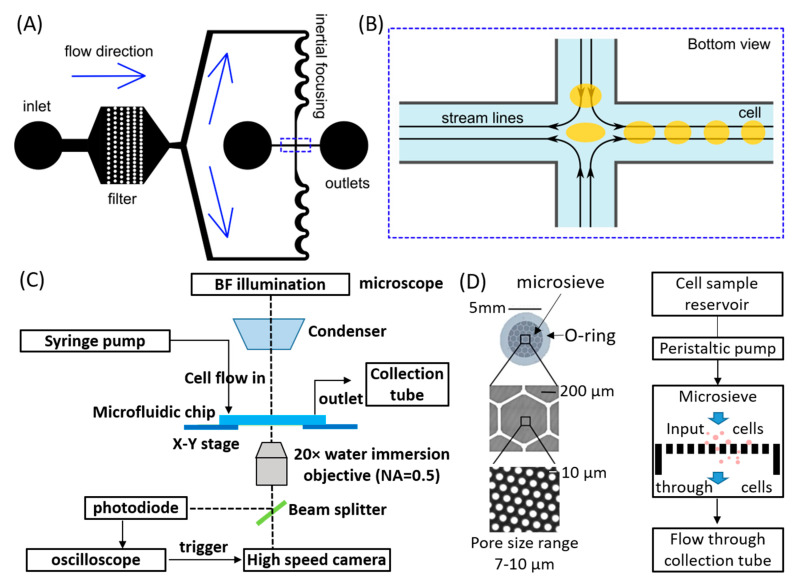
The experimental devices and the working principle. (**A**) A schematic of the individual hydrodynamic stretching microchannel (channel height = 26 μm). The channel consists of one inlet, two outlets and a filtering region to avoid cell aggregates and debris. The flow enters from above and below the central stretching region (marked by the dashed box). Cells are focused to the channel center line by the inertial focusing curvatures and are deformed in the stretching region. (**B**) A sketch of the stretching region, which is marked by the dashed box in A. (**C**) A schematic diagram showing the operation of the hydrodynamic stretching microchannel and image recording. (**D**) The structure of a microsieve and its working principle.

**Figure 2 micromachines-11-00486-f002:**
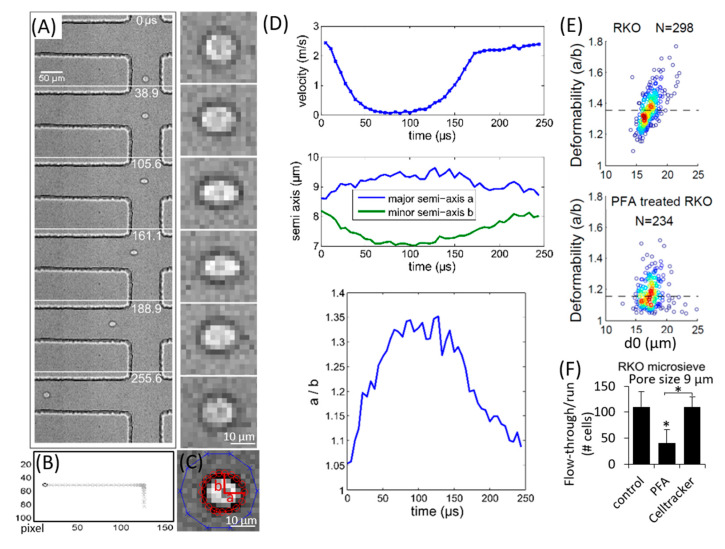
Illustration of image processing and data analysis with an exemplary RKO cell. (**A**) Left column: selected image sequences for a single cell traveling through the stretching extensional flow region. Right column: enlarged images of the cell shown in the left column. (**B**) Cell detection (*circle*) with background subtraction and thresholding, showing trajectory of cell centroid (*cross symbols*) across the channel. (**C**) Active contour method for extraction of cell contour, major, *a*, and minor, *b*, semi-axis. The red circles denote the cell boundary elements detected using the active contour method. (**D**) Extracted data for the temporal evolution of the cell velocity (*top*), its major (*blue*) and minor (*green*) semi-axis (*middle*), and the ratio between the major and minor semi-axis (*a/b*) (*bottom*). (**E**) Statistical analysis of the cell deformability vs. relaxation diameter *d_0_* with the density scatter plot for untreated RKO and paraformaldehyde (PFA)-treated RKO cells. The dashed lines indicate the median deformability. A hotter color indicates a higher data density. The deformability is defined as the maximum value of *a/b*, while *d_0_* is the averaged diameter when the ratio *a/b* is minimum. The PFA-treated RKO cells have a significantly lower deformability compared to untreated RKO cells, *p* < 0.0001 from two-tailed student t test. (**F**) Averaged number of cells flowing through the microsieve (pore size 9 µm) per run for non-treated RKO (control), PFA-treated RKO and RKO loaded with cell tracker fluorescence dye with the same input number of cells. Three replicates were done for each microsieve experiment (n = 3). There are significantly fewer flow-through cells for PFA-treated RKO compared to the control group, * *p* < 0.05 from one-way ANOVA test followed by post-hoc Tukey Honest Significant Difference (HSD) test. No significant difference is observed between control and cell tracker loaded group (*p* = 0.90). The error bars are standard deviations from three repeated microsieve measurements.

**Figure 3 micromachines-11-00486-f003:**
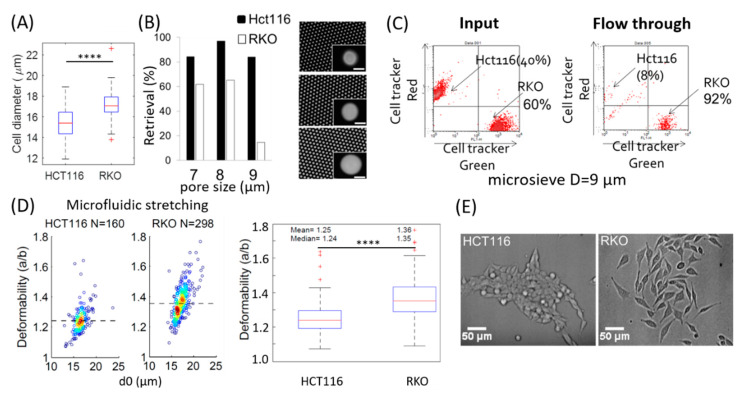
Differential deformability and characteristics of colorectal cancer cell lines HCT116 and RKO. (**A**) Boxplot and statistic results for the diameter of HCT116 and RKO cell lines. Cell line diameters were extracted from images of freshly trypsinized cells in suspension using the software ImageJ. The cell images are taken from an inverted microspore (Olympus IX81) in bright field modus. The line inside the boxes represents the median of the data, while the edges of the boxes are the 25th and 75th percentiles, and the ends of the whiskers denote 1.5 times the interquartile deviation (IQR). The plus symbols on the top or bottom of the whiskers indicate data not included between the whiskers. **** *p* < 0.0001 from two-tailed student t test. (**B**) Retrieval (normalized as the percentage of cells left over on the microsieve surface in total input cells) of HCT116 and RKO cells from microsieves with a pore diameter of 7, 8, and 9 µm (shown on the right side, scale bars denote 7 µm), respectively. (**C**) Percentage of both cell lines in input and after flowing through the microsieve at high flow rate of 1 mL/min and identified with fluorescent cell tracker. (**D**) HCT116 and RKO from microfluidic hydrodynamic stretching: (left) density scatter plot of the deformability. The dashed lines indicate the median deformability. (right) Statistic analysis (the boxes). The HCT116 cells have a significantly lower deformability compared to RKO cells, **** *p* < 0.0001 from two-tailed student t test. (**E**) Bright field images of the cell morphology during adhesion and growth of HCT116 and RKO under the same culture condition.

**Figure 4 micromachines-11-00486-f004:**
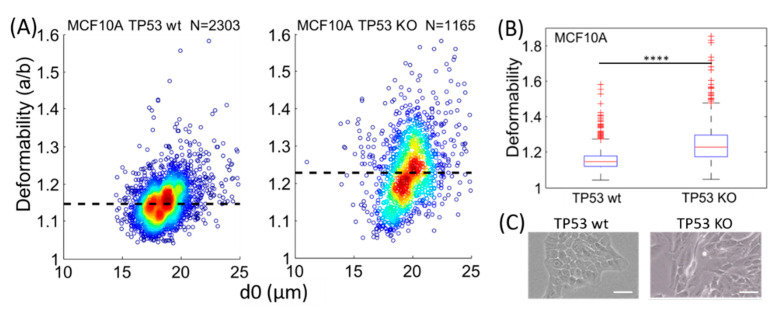
Differential deformability between MCF10A TP53 wildtype (wt) and MCF10A with the knockout of TP53 gene (KO) from the microfluidic hydrodynamic stretching. (**A**) Density scatter plot of the deformability for MCF10A TP53 wt and MCF10A TP53 KO. The dashed lines indicate the median deformability. (**B**) Boxplot and statistical analysis of the above deformability. **** *p* < 0.0001 from two-tailed student t test. (**C**) Bright field images for the cell morphology under the same culture condition. The scale bars denote 20 µm.

**Figure 5 micromachines-11-00486-f005:**
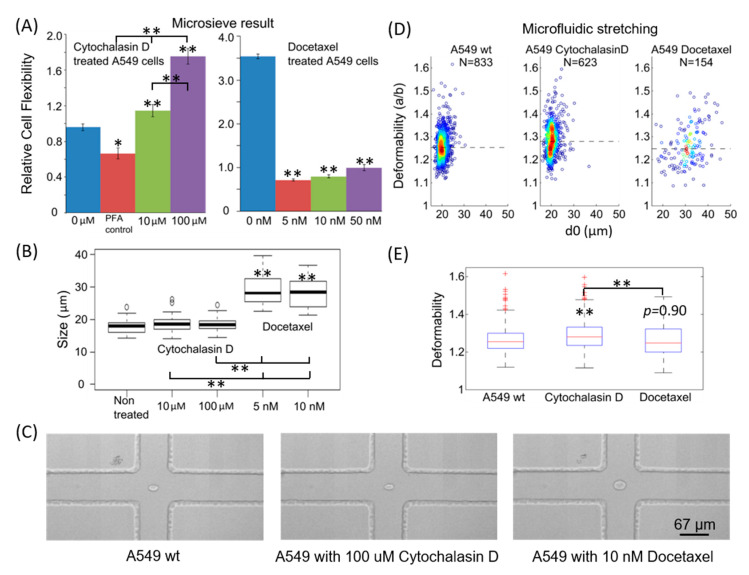
Correlation of cellular deformability of A549 cells with pharmaceutic drug treatment. One-way ANOVA test is performed first to see whether one or more treatment groups are significantly different, followed by post-hoc Tukey HSD test to identify which of the pairs of treatments are significantly different from each other. If not specified, the comparisons are between A549 control (non-treated/wt) and drug treated group, * *p* < 0.05, ** *p* < 0.01. (**A**) Relative cell flexibility from microsieve measurements for A549 control (0 µM), treated with PFA, cytochalasin D (10, 100 µM) and docetaxel (5, 10, 50 nM). (**B**) Statistical analysis for the cell size of A549 non-treated and treated with cytochalasin D (10, 100 µM) and docetaxel (5, 10 nM). Compared to the non-treated group, there is significant increase of the cell size for both 5 and 10 nM docetaxel-treated groups, while no significant change of cell size is found for cytochalasin D-treated groups. (**C**) Representative images of individual cells passing through the center of the stretching region in hydrodynamic stretching microchannel for non-treated A549 (wt) and A549 treated with 100 µM cytochalasin D and 10 nM docetaxel. (**D**) Measured deformability from the microfluidic hydrodynamic stretching for the non-treated A549 and A549 treated with 100 µM cytochalasin D and 10 nM docetaxel: density scatter plot of the deformability. The dashed lines indicate the median deformability. (**E**) Boxplot and statistical analysis of the deformability of (D). A549 treated with cytochalasin D has a significantly higher deformability compared to untreated A549 cells. There is no statistically significant difference between the deformability of docetaxel-treated A549 and the wildtype, *p* = 0.90.

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
