# Peer review of "Single Cell Hydrodynamic Stretching and Microsieve Filtration Reveal Genetic, Phenotypic and Treatment-Related Links to Cellular Deformability"

_micromachines, 2020, doi:10.3390/mi11050486_

Round 1
Reviewer 1 Report
The authors studied cancer cell deformability after EMT and chemotherapy drug treatment by combining the microfluidic hydrodynamic stretching method and microsieve. The analysis itself is nearly flawless however there are some concerns need to be addressed:
(1) For the hydrodynamic stretching and microsieve experiment, what cell concentration (e.g. cells/mL) is used?
(2) For the microsieve method, will the flow rate affect the cell pass-through? How did the authors choose the flow rate?
(3) For the active contour method, what is the advantage in this application compared to other algorithms?
(4) The authors presented the hydrodynamic stretching results as density scatter plots. However, the correlation between cell size and deformability is observed (positive correlation). Will the cell size affect the cell deformability? How to decouple the cell size and deformability, and find the true cell stiffness?
(5) Replicates are lacking for most of the experiments in this manuscript, and the authors should not claim the difference is significant without any statistical test.
(6) Considering the throughput of the hydrodynamic stretching method is very high, why the cell number is low in the scatter plots (only several hundred)?
(7) How robust is the hydrodynamic stretching method? What is the variance of the median deformability for the same cell line? For example, if RKO cells from three different flasks are tested using the hydrodynamic stretching method, will the median deformability change?
(8) In Fig.3B, when pore size increased to 9 um, why the percentage of retrieval decreased compared to 8 um pore size?
(9) Fig.3B and Fig.3C seem contradictory. From Fig.3B, Hct116 cells can pass through the microsieve easily compared to RKO cells, but Fig.3C tells a different story. Please explain.
(10) From my experience, there are a lot of clumpy HCT116 cells even if they are trypsinized, how did the authors make sure that they were measuring single cells?
(11) In Fig.5A, why the relative cell flexibility increased when the concentration of docetaxel increased?
In addition, there are some other comments:
(1) Line 31, “drug docetaxel leads to an increase in the size and deformability” is not true, since only the size increased after treated with docetaxel, according to Fig.5.
(2) In the introduction part, the authors failed to mention RT-DC (Otto et al. Nature Methods, 2015) and iMCH (Deng et al. Small, 2017), which are also important high-throughput techniques for cell deformability measurement.
(3) Fig.1D needs scale bar.
(4) In Fig.2C, what are the red circles?
(5) Line 170, “see Fig. 2C” should be Fig.2E.
(6) In Fig.2F, what does the error bar stands for? Please also check other figures.
(7) line 323, “The increase of cell size is probably because cells get stuck just before mitosis in the cell cycle when treated with Docetaxel” needs references.
Reviewer 2 Report
Overall, this is a nicely written paper with significant experimental aims for the field, but there are some details missing from the methods section and the paper completely lacks the necessary statistical analysis for me to recommend immediate publication. Once proper statistical analysis is done, if the findings which are claimed to be statistically significant are actually statistically significant, it will be a significant contribution to the field.
Methods:
Please include a brief description of the original channel patterning method. Was this standard SU8-based photolithography, micromilling, or some other patterning method that you are then molding from? One sentence on the bonding method used will also help the chip fabrication methods be more complete and reproducible.
Figure 1 caption mentions "dirt" in the cell suspension, this should probably be stated as "debris" since "dirt" should not be in the cell suspension or chips if prepared and cleaned properly.
The concentration of celltracker dyes used should be stated, as at too high of concentrations these can sometimes change cellular behavior (and potentially deformability). Please also include culture conditions/media for all cell lines, or clearly state that all cells in all experiments were grown in the same conditions.
Are there limits to cell concentration that could affect the performance of the chip? If so the cell concentrations used should be clearly stated.
State statistical analysis to be used in methods section. Also state if plotted error bars are standard deviation or standard error mean.
Results:
"The PFA treated RKO cells have a significant lower deformability" - need to include p-value to claim any significant difference.
"fewer PFA treated cells can pass through the microsieve than the 221 non-treated (control) ones" - also need to do statistical analysis and include p-value with this claim. Were replicates done for each microsieve experiment or are these single tests (n=?)
Claims for significant differences for HCT116 and RKO data also need statistical analysis beyond simply plotting box and whisker plots (include p-value).
Claims for significant differences for A549 and different treatment effects on deformability also need p-values included.
Once statistical analysis is done, significant differences should be noted on figures.
Discussion:
Claims of differences in deformabilitiy/flexibility should not be made without proper statistical analysis. While the results to have the potential to make a significant contribution to the field, the "significant" differences claimed to be found between treatment groups must first be validated and then reported appropriately.
Round 2
Reviewer 1 Report
The authors have addressed all my concerns, and I would recommend the paper to be accepted for publication.
Author Response
Many thanks to the reviewer for the support of our study and revisions.
Reviewer 2 Report
The authors adequately responded to most of the review comments, although I still think the statistical analysis is inappropriate in certain sections.
For simply comparing only two groups, the t-test is adequate and presented appropriately. However, in cases like Figure 5, where more than 2 groups are being compared to each other, an ANOVA test should be used to test for significance first, followed by a posthoc multiple comparisons test to determine p-values between treatment groups. This is a straightforward comparison that can be easily performed in any statistics software and will avoid the additive error inherent in doing multiple pairwise t-tests between each group independently within a single experiment.
